

# *Halimeda tuna* (Bryopsidales, Ulvophyceae) calcification on the depth transect in the northern Adriatic Sea; carbonate production on the microscale of individual segments

Yvonne Nemcova[1], Martina Orlando-Bonaca[2] and Jiri Neustupa[1]

[1] Department of Botany, Faculty of Science, Charles University, Prague, Czech Republic
[2] Marine Biology Station Piran, National Institute of Biology, Piran, Slovenia

## ABSTRACT

*Halimeda tuna* (J. Ellis & Solander) J.V. Lamouroux is the only *Halimeda* species found in the Mediterranean Sea, and it is an important habitat former. In the northern Adriatic, *H. tuna* is among the ten most abundant seaweeds in the upper-infralittoral belt in spring and autumn. The modular thalli consist of serially arranged calcified segments. Calcification is closely related to photosynthesis, which causes alkalinization of the inter-utricular space and triggers aragonite formation. Understanding of the complex patterns of segment shape plasticity in relation to $CaCO_3$ content at different depth levels is still incomplete. Geometric morphometrics was used to investigate *H. tuna* segment shape variation on the depth transect at Cape Madona Nature Monument in the northern Adriatic Sea. The position on the thallus and the $CaCO_3$ content of each studied segment were recorded, allowing slight changes in mineral content to be detected at the microscale of the segments. Our results showed that shape, size, or asymmetry of *H. tuna* segments were not significantly affected by depth. On the other hand, plants that grew deeper were generally more calcified. The apical and subapical segments contributed to the increase in $CaCO_3$ content at the deeper sites, whereas the basal segments did not. This indicates that reniform or oval segments positioned apically or subapically play a key role in calcification of *H. tuna* in Mediterranean ecosystems.

## INTRODUCTION

The species of the calcifying genus *Halimeda* J. V. Lamouroux (Bryopsidales, Halimedaceae) are important primary producers (*Hillis-Colinvaux, 1980*). They are abundant especially in tropical shallow seas and are major contributors to inter-reef carbonate deposits (*McNeil et al., 2016*). *Halimeda* mud-sustained bioherms along the northern Great Barrier Reef represent important habitats and potential carbon sink; their contribution to the sediment budget may equal or exceed the production of carbonate sediments by stony corals (*McNeil et al., 2016*; *McNeil et al., 2022*). By fixation and long-term storage of atmospheric carbon

Corresponding author
Yvonne Nemcova,
ynemcova@natur.cuni.cz

dioxide in tropical and subtropical reef environments, *Halimeda* provides important ecosystem services (*Mayakun & Prathep, 2019*). Although *Halimeda* spp. are not physical habitat engineers like *e.g.*, calcareous red algae, they effectively contributed to lime muds and sands in the Phanerozoic (*Basso & Granier, 2012*; *Granier, 2012*). *Halimeda tuna* (J. Ellis & Solander) J.V. Lamouroux is the only species thriving in the Mediterranean Sea and is one of the most important biogenic carbonate producers (*Lipej, Orlando-Bonaca & Mavrič, 2016*). In the Adriatic Sea, the community dominated by *H. tuna* and the red alga *Mesophyllum alternans* (Foslie) Cabioch & M.L.Mendoza produced about 465 g CaCO$_3$ per m$^2$ per year (*Canals & Ballesteros, 1997*). Assemblages of *H. tuna* thrive in relatively shallow waters. *Halimeda tuna* may also occur in the Mediterranean Coralligenous formations, a complex of predominantly calcareous-invertebrate biocenoses with important concretion potential, so *Halimeda* fragments may enter the sedimentary record (*Basso et al., 2022*). In the northern Adriatic, it is also among the ten dominant seaweeds in terms of abundance in spring and autumn samples collected at reference sites for macroalgae in the upper-infralittoral belt (*Orlando-Bonaca & Rotter, 2018*). According to the Ecological Evaluation Index continuous formula (*Orfanidis, Panayotidis & Ugland, 2011*), a multimetric scale-based biotic index that reveals the response of benthic macrophytes to anthropogenic stress, *H. tuna* is included in the first Ecological State Groups (ESG I) with other perennial species, that are indicators of good/high ecological status of coastal waters (*Orlando-Bonaca, Pitacco & Lipej, 2021*).

Halimeda thalli are composed of calcified segments separated by non-calcified nodes. The entire thallus consists of a single branched cell (siphonous thallus). Linear arrays of segments are strung together by medullary siphons that branch to form peripheral utricles (*Hillis-Colinvaux, 1980*). Segments are added consequently, and their production is seasonal, reaching its maximum in summer. However, age of adjacent segments can vary greatly as individual segments and entire segment branches are shed regularly, usually due to physical disturbance and herbivory (*Drew, 1983*; *Ballesteros, 1991*). The peripheral (primary) utricles adhere to each other and form a closed inter-utricular space (IUS) in which aragonite microcrystals precipitate (*Wilbur, Hillis & Watabe, 1969*; *Borowitzka & Larkum, 1977*). The size and shape of peripheral utricles, which have a honeycomb structure when viewed from above, vary among species and represent one of the important identification features (*Borowitzka & Larkum, 1977*; *Peach et al., 2017*). In addition to among-species differences, variation in size and symmetry of peripheral utricles within a single segment of *H. tuna* has also been demonstrated. The utricles near the segment bases were considerably smaller than those located near the apical and lateral margins, and symmetry of the utricles decreased from the centre to the margins (*Neustupa & Nemcova, 2020*).

The inter-utricular space represents a boundary layer that mediates the interaction of algal metabolism with the carbon chemistry of seawater, resulting in CaCO$_3$ precipitation. Calcification is closely linked to photosynthesis, which causes alkalinization of the inter-utricular space and is a trigger for aragonite formation (*Borowitzka & Larkum, 1987*; *McNicholl, Koch & Hofmann, 2019*). However, this process cannot be considered a simple abiotic precipitation because many polysaccharides, proteins, and enzymes are involved

(*Wizemann, Meyer & Westphal, 2014*). The function of calcification is still matter of debate. It has been suggested that calcification acts as a mechanical support in benthic marine environment, provides carbon dioxide and protons for photosynthesis and nutrient uptake, limits grazing, and prevents attack by parasites and viruses (*Schupp & Paul, 1994*; *Raven & Giordano, 2009*). The extent of calcification is species-specific; species living in the high-energy regimes usually have smaller and heavily calcified segments (*Multer, 1988*; *Kooistra & Verbruggen, 2005*). In general, calcification rate in *Halimeda* spp. is influenced by rate of photosynthesis, light, and nutrient availability (*Vroom et al., 2003*; *Yniguez, McManus & Collado-Vides, 2010*; *Pongparadon, Nooek & Prathep, 2020*).

The shape of the segments can be a relatively plastic feature (*Vroom et al., 2003*; *Pongparadon, Nooek & Prathep, 2020*). *Verbruggen et al. (2005)* evaluated the utility of segment size and shape, assessed by geometric morphometrics and elliptic Fourier analysis, as diagnostic features in *Halimeda* specimens representing the five lineages of the genus delimited by molecular data. Segment size and shape were found to be relatively good predictors of species membership, with the exception of deviated segments (*Verbruggen et al., 2006*). These are apical, non-calcified segments and those from the basal part of the thallus that deviate from the species-typical shape. Plasticity of segment shape in response to environmental factors such as light availability (*Neustupa & Nemcova, 2018*; *Pongparadon, Nooek & Prathep, 2020*) or wave exposure (*Littler & Littler, 2000*; *Kooistra & Verbruggen, 2005*; *Pongparadon, Nooek & Prathep, 2020*) has also been described.

So far, the most detailed description of *H. tuna* segment shape and size plasticity was provided by *Neustupa & Nemcova (2018)*, who studied two populations from the northern Adriatic Sea. A specific developmental pattern of *Halimeda tuna* was revealed. Shape and size of the segment were strongly determined by its position on the thallus. Basally located segments were generally smaller and inversely conical in shape, while segments in the middle and upper part of the thallus were large, oval to reniform in shape. Although both populations were sampled from the same depth in the upper sublittoral, there was a significant difference in the mean shape of the segments, indicating that local environmental factors may have influenced segment morphogenesis (*Neustupa & Nemcova, 2018*). Later, *Neustupa & Nemcova (2022)* examined size and shape plasticity, and calcium carbonate content of *H. tuna* segments in populations along the Adriatic latitudinal gradient. They showed, similarly to the previous study, that segment position on the thallus was the main determinant of its shape. The effect of position outweighed shape differences among plants, populations, and regions. In addition, segment shape proved to be a significant predictor of its $CaCO_3$ content. The reniform and oval segments contained significantly more calcium carbonate than the segments of conversely conical shape.

Depth represents a complex environmental variable; habitats on the depth gradient tend to vary in light and nutrient availability, the extent of disturbance, and may undergo varying pressures from herbivores. According to some field observations, thalli of *Halimeda* spp. are more heavily calcified in deeper waters than in shallower ones (*Goreau, 1963*); however, this effect was less pronounced in *H. tuna* (*Böhm, 1973*). On the other hand, calcification rate and $CaCO_3$ content have been described to decrease at low light intensities, which normally correspond to deeper habitats (*Borowitzka & Larkum, 1976*; *Bandeira-Pedrosa, Pereira &*
*Oliveira, 2004*; *Pongparadon, Nooek & Prathep, 2020*). Reduced calcification performance observed in *H. tuna* populations under extremely high light irradiance in shallow Florida Keys reef systems was explained by photoinhibition (*Vroom et al., 2003*). *Halimeda* plants growing in shallow, well-irradiated habitats often developed smaller segments than those growing in deeper, shaded localities (*Vroom et al., 2003*; *Bandeira-Pedrosa, Pereira & Oliveira, 2004*; *Pongparadon, Zuccarello & Prathep, 2017*; *Pongparadon, Nooek & Prathep, 2020*). Information on depth-dependent segment shape plasticity remains fragmentary (*Pongparadon, Nooek & Prathep, 2020*). Understanding complex patterns of segment-shape plasticity in relation to CaCO$_3$ content at different depth levels may provide an indication of how individual plants/segments contribute to the overall carbonate budget.

Here we present a detailed analysis of *H. tuna* segment shape and size variation on the depth transect at Cape Madona Nature Monument (NM) (northern Adriatic Sea) in relation to calcium carbonate content. The percentage of CaCO$_3$ content was determined for each individual segment. A landmark-based geometric morphometric framework was used to analyze shape and size of the segments. We hypothesize that segments increase in size and calcium carbonate content with increasing depth. We also question whether all segments contribute equally to depth-related calcium carbonate content change.

## MATERIALS & METHODS

### Sampling

The Gulf of Trieste is an epicontinental, shallow, semi-enclosed sea basin in the northernmost part of the Adriatic Sea with an average depth of almost 21 m. The Gulf expands from Cape Savudrija (Croatia) to Grado (Italy) and includes the entire Slovenian coast. This area has the lowest winter temperatures in the Mediterranean, usually below 10 °C. The prevailing winds blow mainly from the northeast in an offshore direction (*Boicourt et al., 2021*). In the bottom layer, below 10 m depth, water circulation is mostly counter-clockwise, while at the surface, in a layer about 5 m thick, circulation is driven clockwise by the wind (*Stravisi, 1983*). The coastal zone has been affected by various anthropogenic impacts in recent decades, such as construction, intensive fishing, sewage discharges and mariculture (*Orlando-Bonaca et al., 2015*). However, according to the TRIX index (a combination of loads and impact indicators), water quality conditions of Slovenian coastal waters were recently evaluated as elevated (*Giovanardi et al., 2018*).

*Halimeda tuna* thalli were sampled on October 13, 2021, along the depth transect at Cape Madona NM, the westernmost part of the Piran peninsula, Slovenia (45°31′50″; 45°31′50″) by means of scuba-diving and snorkelling. The Cape Madona NM met the selection criteria for macroalgae reference sites in Slovenian coastal waters (*Orlando-Bonaca & Rotter, 2018*). It is located in the Slovenian water body Sl5VT4, where a large part of the coastal belt is still in its natural state (*Orlando-Bonaca, Lipej & Orfanidis, 2008*). A marked decline in the total coverage of canopy-forming algae, especially for *Cystoseira s.l.* spp., was reported for the macroalgal sampling site at Cape Madona NM in the last decade (*Orlando-Bonaca, Pitacco & Lipej, 2021*).

At each depth level (0.5, 2.0, 4.0, 4.9, 6.2 and 9.2 m) ten plants were sampled from 2 m×2 m quadrats and light intensity at the seafloor was measured by LI-COR Underwater

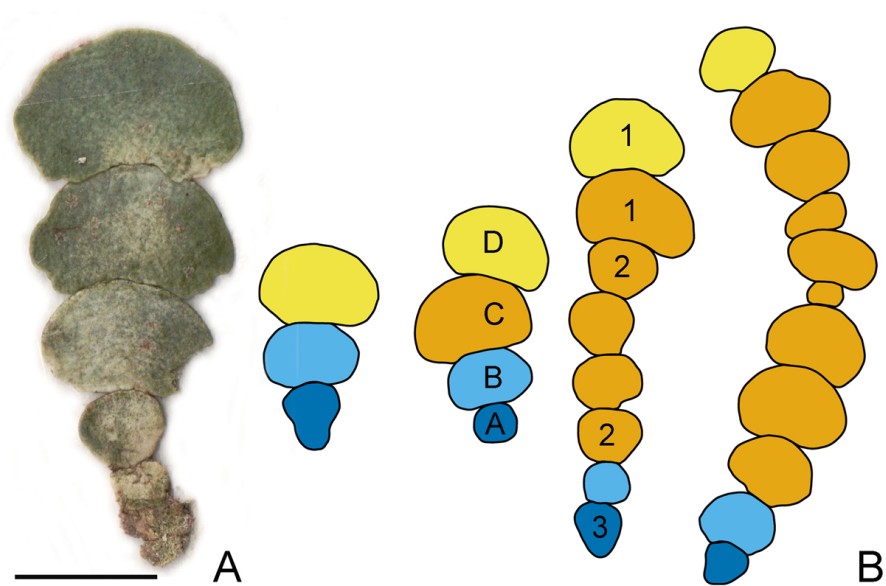

**Figure 1** *Halimeda tuna* **thallus and designation of the segment positions.** (A) *H. tuna* thallus consisting of seven segments; bar = 1 cm. (B) Designation of the segment positions on the thallus of *H. tuna*: A-basal (dark blue), B-suprabasal (light blue), C-subapical (ochre), D-apical (yellow). Shapes of *H. tuna* segments: 1- reniform, 2-oval, 3-inversely conical. Note that the segments of irregular shape may also be present.

Quantum Sensor Model Number: LI-192 (90.0, 55.0, 31.0, 29.0, 16,3 and 10.3 $\mu$mol m$^{-2}$ s$^{-1}$, respectively) at 10:10 a.m. In the laboratory, thalli were cleaned from the epiphytes, pressed on absorbent paper, and scanned. The segments composing the longest available series within each plant were used for further analyses, occasionally occurring uncalcified apical segments with incomplete morphogenesis were eliminated. The dataset consisted of 379 segments from 60 different plants comprising the same population. The longest available series of segments vary from three to 12 segments per plant (average 6.3). Segment positions were labelled: A-basal, B-suprabasal, D-apical (in case of three segments) in longer branches (four and more segments) all other segments were labelled C-subapical (see Fig. 1).

## Mineral content of segments

Segments of the longest available series (composing the longest branch), were carefully separated under the dissecting stereomicroscope, dried at 60 °C for 45 min and weighed using a precise balance XPR6UD5/M Mettler-Toledo GmbH, Switzerland. Individual segments were decalcified in 5% HCl for 10 min until bubbling had ceased (*Vroom et al., 2003*), rinsed in deionized water, dried (60 °C for 45 min) and weighed again. The proportional mineral content of each segment was determined as previously described in *Neustupa & Nemcova (2022)*. $C = 1 - dcw/drw$, where *drw* is the total dry weight and *dcw* is the dry weight after decalcification.

## Geometric morphometrics

The outlines of individual segments were registered by a series of 80 equidistant two-dimensional points located along their margins. In each segment, the single fixed point (landmark) was located at the position of the basal node. Then, the remaining 79 points were placed in equidistant positions along the segment outlines and treated as semilandmarks in subsequent analyses (*Zelditch, Swiderski & Sheets, 2012*). To minimize potential effects of digitization error that could affect the analysis of shape asymmetry in particular, each segment was registered twice in clockwise and counter-clockwise direction. Subsequently, the counter-clockwise registered points were relabeled to match the order of clockwise digitization. The averaged coordinates from both digitizations were then used for subsequent analyses. The landmark and semilandmark coordinates of segment outlines are listed in Table S1. For the analysis of shape variation among segments, their symmetrical halves were averaged to remove the effects of bilateral asymmetry. In this procedure, each segment was reflected along the axis of bilateral symmetry. Then, the points in the reflected copy were relabeled to match their designation in the original configuration. Finally, the coordinates of the original and mirrored copies were averaged to obtain the ideally symmetric configuration of each segment (*Klingenberg, 2015*).

Subsequent shape analyses were based on the coordinates yielded by the generalized Procrustes analysis (GPA), which minimizes the sum of squared distances among corresponding landmarks by removing extraneous information of size, location and orientation (*Zelditch, Swiderski & Sheets, 2012*; *Bookstein, 2018*). Furthermore, the GPA included an additional step consisting of iterative sliding of the semilandmarks along the outline tangents so that their final position resulted in the smoothest possible deformation of the actual configuration relative to the mean shape of the dataset under study (*Bookstein, 1997*).

Outline points were digitized using the semi-automated *background curves* tool of TpsDig, ver. 2.22 (*Rohlf, 2015*). GPA with sliding of semilandmarks was implemented in TpsRelw, ver. 1.65. Configurations of shapes positioned at the margins of the ordination space were illustrated using TpsRelw, ver. 1.65. Similarly, segment shapes typical of the minimum and maximum values of independent factors reconstructed by the multivariate regression models were illustrated by TpsRegr, ver. 1.50 (*Rohlf, 2015*).

## Statistical analyses

Multivariate patterns of shape variation were illustrated by principal component analysis (PCA) of coordinates aligned by GPA. Linear correlations between external factors and individual PCs were shown as vectors in the ordination plot. In addition, the significance of these relationships was evaluated by permutation tests on Pearson's *r* and Spearman's non-parametric rank-order correlation coefficients.

The effects of various external factors on the shape characteristics of the segments were evaluated by a series of parallel multivariate regression models. These analyses evaluated the amount of shape variation among configurations, expressed by the Procrustes sum of squares (SS), that could be attributed to each factor. The probabilities of the null hypotheses, which assume that there are no relationships between shape variation among

segments and external factors, were assessed by comparing these measured Procrustes SS values with the distribution of random SS yielded by 999 permutations of the original data (*Anderson, 2017*; *Schaefer et al., 2006*). The Bonferroni correction for multiple comparisons was used to set the appropriate limit on the significance of individual models (*Perrett & Mundfrom, 2010*).

A total of eight external factors were tested. Depth values spanned six levels according to the sampling design. Mineral content was assessed as proportion of $CaCO_3$ in the dry weight of the segments. Centroid size of the segments, which is defined as the square root of the sum of squared distances of each point from the centroid of their respective configurations, was used as a size measure to assess the shape allometry within the dataset. To assess the relationship of centroid size with segment surface area, surface area was determined using the *curve area* tool of TpsDig, ver. 2.22 (*Rohlf, 2015*). Centroid size (Table S2) was closely linearly correlated with the surface area of the segments (Pearson's $r = 0.98$), so we used only centroid size for further analyses. Bilateral shape asymmetry was quantified by Procrustes distances between the original and mirrored configurations of each segment. Finally, four binary variables were created to identify the different positions of the segments within the thalli. The relationships among these external factors were evaluated by Spearman's non-parametric rank-order correlation analyses. In addition, Spearman's rank-order correlation was also computed for the relationships of depth and mineral content of the segments at each of the four positions within the thalli. Bonferroni-corrected significance of these correlation analyses was assessed by 999 permutations.

PCA of the shape data was performed using TpsRelw, ver. 1.65 (*Rohlf, 2015*). The multivariate regression models were implemented using the function *procD.lm* of the package geomorph, ver. 4.0.0 (*Baken et al., 2021*) in R, ver. 4.0.5 (*R Core Team, 2021*).

## RESULTS

In the PCA ordination plot the shape data of 379 objects (segments) are represented by points within a morphospace spanned by the first two PCs (Fig. 2). Most of the shape variation (77.6%) was spanned by PC1, which illustrated the shape changes between the broadly reniform segments in the negative part of the axis and the inversely conical segments on the opposite side of PC1. The second most important trend was represented by PC2 (8.9% of shape data variation). This axis illustrated differences between segments with a narrow base and those with more concave outlines. Segment shapes typical for the most marginal occupied positions along these two PC axes are outlined. Segment size differences are not shown in the outlines as the shape variables were size standardized prior to analysis. Size is treated as one of the independent factors projected as vectors onto the morphospace (Fig. 2). The prominent effect of the segment position within the thalli (blue vectors) was clearly demonstrated. Positions "A" and "B" (basal and suprabasal segments) were associated with the right part of the morphospace typical of the inversely conical segments, whereas the segments in positions "C" and "D" (subapical and apical segments) were located in the left part, indicating that these positions contained predominantly reniform segments. Size was represented by a very long vector pointing to the left side

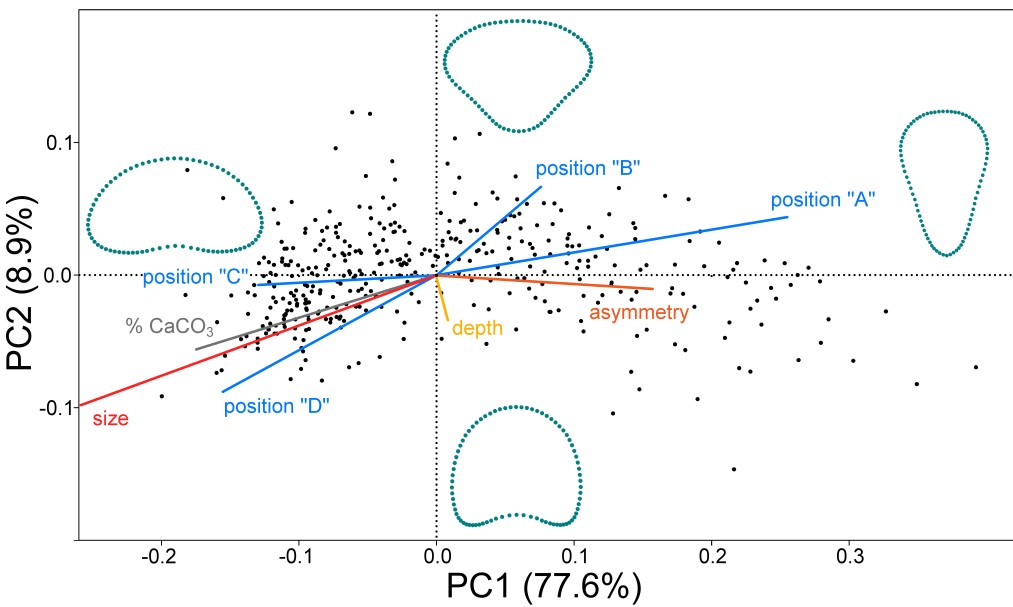

**Figure 2** **PCA ordination plot of the segment shape data showing first two PCs (PC1 *vs.* PC2).** The configurations illustrate shapes typical for the most marginal occupied positions on each PC. Independent factors (depth, size, calcium carbonate content, asymmetry, and the positions of the segment along the branch A, B, C and D) are projected as vectors onto the morphospace.

of the morphospace. Size was also positively correlated with positions "C" and "D" and the percentage of calcium carbonate content in the segments (grey vector). On the other hand, more asymmetric segments were located in the right part of the morphospace, predominantly occupied by the inversely conical segment shapes in positions "A" and "B" (Fig. 2). Thus, the reniform segments located subapically and apically on the thalli (positions "C" and "D") were larger, more calcified, and more symmetrical compared to the inversely conical segments at the base of the thalli (position "A"), which tended to be smaller, less calcified, and more asymmetrical. Interestingly, depth (green vector) was not significantly correlated with either PC1 or PC2 (Table 1).

The multivariate regression models largely confirmed the PCA results. Segment size proved be the most important independent factor, explaining nearly 43% of the shape variation (Table 2). While comparatively smaller segments were clearly typical of inversely conical shapes, increasing size was associated with reniform shapes (Fig. 3A). A very similar shape change was also associated with the varying mineral content of the segments. The reniform segments generally had higher $CaCO_3$ content then the conical-shaped ones (Fig. 3B). However, this regression model accounted for only about 10.8% of the total shape variation. The position of the segments on the thalli proved to be significantly related to their shape (Table 2, Figs. 3C–3F). The strongest relationship was found for segments located at the base of the thalli (position "A"), which are typical of inversely conical narrow shapes (Fig. 3C). This multivariate regression model explained 21.4% of the total shape variation among segments. In contrast, the subapical and apical segments (positions "C"

Nemcova et al. (2023), *PeerJ*, DOI 10.7717/peerj.15061

**Table 1   Results of linear correlation analyses between principal components yielded by PCA of shape data and external factors.** For each relationship Pearson's correlation and Spearman's non-parametric rank-order correlation coefficients are depicted. Significant Bonferroni-corrected $p$-values lower than 0.0016 are depicted in bold.

| | eigenvalue | % variance | Pearson's $r$ / Spearman rank-order linear correlation | | | | | | | |
|---|---|---|---|---|---|---|---|---|---|---|
| | | | depth | centroid size | % CaCO$_3$ | asymmetry | position "A" | position "B" | position "C" | position "D" |
| **PC1** | 0.01253 | 77.55 | 0.010/−0.077 | **−0.736/−0.782** | **−0.368/−0.379** | **0.330/0.329** | **0.523/0.468** | 0.146/**0.182** | **−0.259/−0.216** | **−0.303/−0.339** |
| **PC2** | 0.00143 | 8.86 | −0.086/−0.116 | **−0.282/−0.285** | −0.126/−0.117 | −0.024/0.025 | 0.091/0.056 | 0.124/0.145 | −0.020/−0.003 | **−0.176/−0.185** |
| **PC3** | 0.00089 | 5.48 | −0.025/−0.031 | **−0.179**/−0.136 | −0.048/−0.006 | 0.098/0.094 | 0.071/0.053 | −0.046/−0.042 | −0.032/−0.044 | 0.015/0.046 |
| **PC4** | 0.00049 | 3.08 | **−0.253/−0.289** | −0.001/−0.021 | −0.029/−0.035 | 0.015/0.014 | −0.016/−0.022 | −0.085/−0.062 | 0.104/0.112 | −0.048/−0.075 |

**Table 2  Results of permutational multivariate analyses of variance relating variation in segment shapes to different independent factors.** Significant Bonferroni-corrected $p$-values lower than 0.0063 are depicted in bold.

| | $\sum PD_{\text{ref}}^2$ | $\sum PD_{\text{resid}}^2$ | $\sum PD_{\text{pred}}^2$ | % explained | % unexplained | $p$ |
|---|---|---|---|---|---|---|
| shape ∼depth | 6.257 | 6.235 | 0.022 | 0.347 | 99.653 | 0.225 |
| shape ∼centroid size | 6.257 | 3.567 | 2.690 | 42.986 | 57.014 | **0.001** |
| shape ∼%CaCO₃ | 6.257 | 5.583 | 0.674 | 10.773 | 89.227 | **0.001** |
| shape ∼asymmetry | 6.257 | 5.699 | 0.558 | 8.917 | 91.083 | **0.001** |
| shape ∼position "A" | 6.257 | 4.915 | 1.342 | 21.444 | 78.556 | **0.001** |
| shape ∼position "B" | 6.257 | 6.141 | 0.116 | 1.857 | 98.143 | **0.006** |
| shape ∼position "C" | 6.257 | 5.925 | 0.332 | 5.312 | 94.688 | **0.001** |
| shape ∼position "D" | 6.257 | 5.790 | 0.467 | 7.466 | 92.534 | **0.001** |

**Notes.**
$\sum PD_{\text{ref}}^2$, sums of squared PDs between each specimen and the reference.
$\sum PD_{\text{resid}}^2$, sums of squared PDs between each specimen and its predicted configuration.
$\sum PD_{\text{pred}}^2$, sums of squared PDs of predicted fit.
% explained, proportion of the total shape variation explained by the multivariate regression model.
% unexplained, proportion of the residual shape variation.
$p$, probability of the null hypothesis based on comparison of the original $\sum PD_{\text{pred}}^2$ with the distribution of the 999 random values yielded by the permutations of specimens among the values of the independent factors.

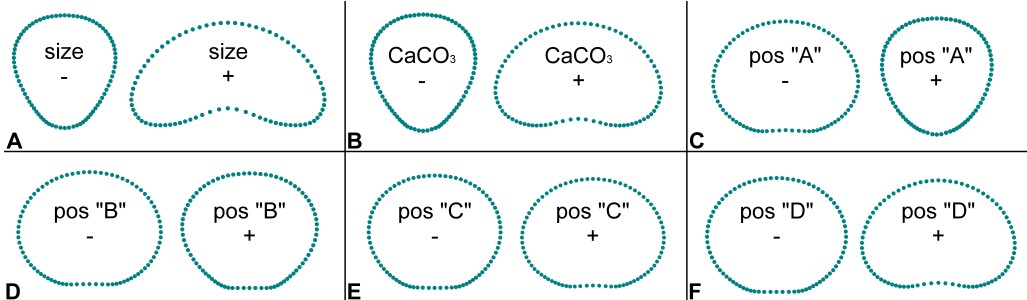

**Figure 3  Patterns of shape change in segments of *Halimeda tuna* related to independent factors and their position on thalli.** The depicted shapes were reconstructed by a series of multivariate regressions of segment shape data on the six independent factors (size, CaCO3 content, thallus position A, B, C, D). Configurations depict segment shapes typical for marginal positions along the shape trajectories within the observed variability of the studied dataset.

and "D") typically exhibited the reniform shape (Figs. 3E–3F). Depth proved to be virtually independent and unrelated to segment shape and accounted only for a negligible 0.3% of the variation (Table 2).

Similarly, depth was largely independent of segment size and their shape asymmetry (Fig. 4). On the other hand, a significant positive relationship was found between depth and mineral content with the non-parametric Spearman's $r_s = 0.21$. This means that plants growing in deeper locations (6–9 m) have slightly more calcified segments than those growing closer to the sea surface. This relationship proved to be even more pronounced for subapical and apical segments in positions "C" ($r_s = 0.33$) and "D" ($r_s = 0.47$). Thus, the segments in the apical position showed the strongest correlation. In contrast, the mineral content of the basal segments in positions "A" and "B" was not significantly related to

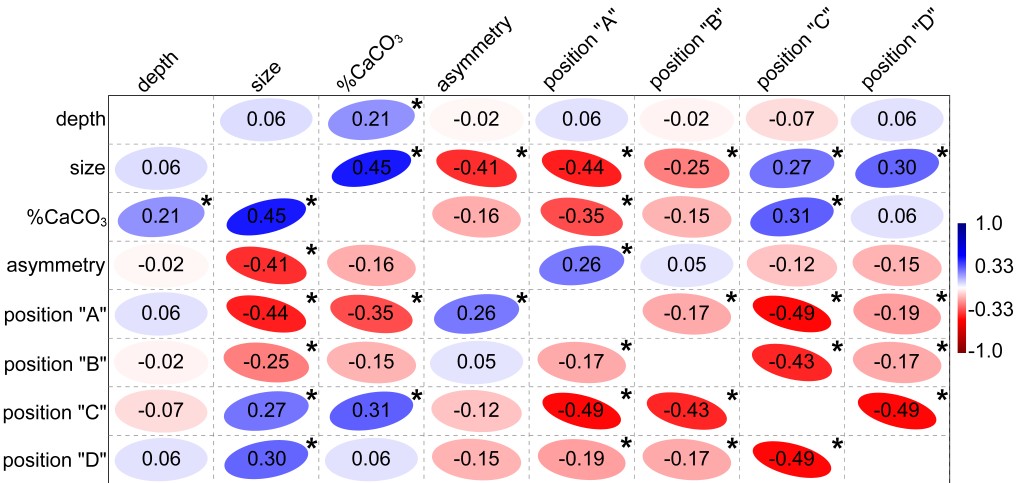

**Figure 4** Correlation plot showing Spearman's non-parametric rank-order coefficients among tested independent factors. The positive correlation is shown in blue, the negative correlation in red. Significant Bonferroni-corrected *p*-values lower than 0.0018 are marked by asterisks.

depth, with $r_s = -0.06$ and $r_s = 0.14$, respectively. The basal segments in position "A" had a significantly lower mineral content with $r_s = -0.35$. On the other hand, the subapical segments in position "C" were typical of a comparatively higher CaCO$_3$ content ($r_s = 0.31$). The suprabasal (position "B") and apical segments (position "D") were not significantly related to mineral content (Fig. 4). The relationship between the different positions and centroid size largely confirmed the pattern established by PCA. Positions "A" and "B" were significantly negatively related to segment size, while the opposite relationship was found for segments located in positions "C" and "D" (Fig. 4).

CaCO$_3$ content increased significantly with segment size (Figs. 4 and 5). The percentage increase in CaCO$_3$ content along the regression line was 15–20%. In addition to the generally smaller basal and suprabasal segments (position "A" and "B"), some of the subapical segments (position "C") were also relatively less calcified. In general, the mineral content of the subapical segments proved to be the most plastic in relation to varying size (Fig. 5). However, even the smallest, most basal segments had a significant amount of CaCO$_3$, although the percentage was much lower compared to the large reniform segments (approximately 30% *vs.* 87% CaCO$_3$ content).

## DISCUSSION

The marine green macroalga *Halimeda* is an important calcifying organism contributing significantly to the calcium carbonate budget of marine ecosystems (*Hillis-Colinvaux, 1980*; *McNeil et al., 2016*). We still have limited information on calcification performance as a function of depth. Several studies aimed to compare the overall thallus morphology, segment size, and general shape, and to assess CaCO$_3$ content of the whole thalli, usually at two geographically close localities of different depths (*Vroom et al., 2003*) or at a larger number of geographically distant sites (*Pongparadon, Nooek & Prathep, 2020*). In our study

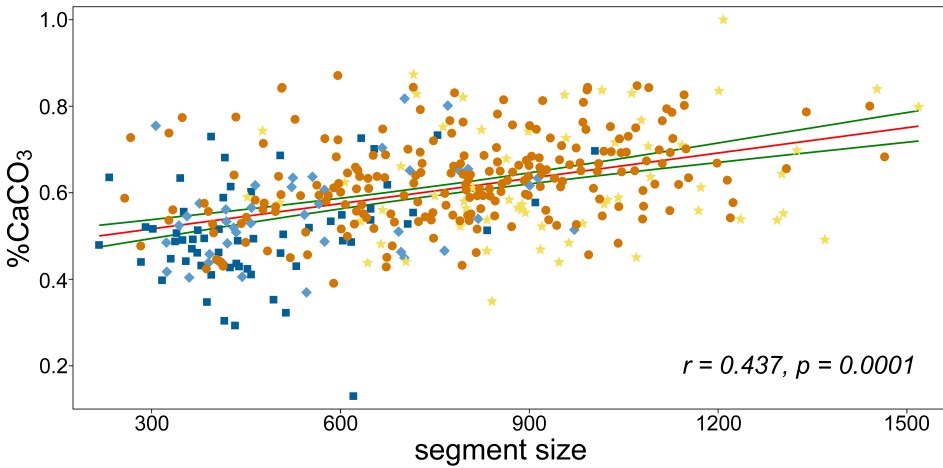

**Figure 5** **The linear correlation analysis of CaCO$_3$ content and segment size.** Pearson's r and the corresponding permutation *p*-value are depicted. Green curves indicate the 95% confidence intervals of the regression line. Colors correspond to positions of segments on branches as illustrated in Fig. 1B.

we provided a detailed analysis of *H. tuna* segment series, harboring plants on a depth transect comprising six depth levels within the same area. Instead of linear measurements, we applied extremely powerful geometric morphometrics to record and statistically evaluate segment size and shape. In addition, we recorded the position on the thallus and CaCO$_3$ content of each examined segment. Thus, we were able to track slight changes in mineral content on the microscale of the segments. Compared to our previous studies (*Neustupa & Nemcova, 2018*; *Neustupa & Nemcova, 2022*), the *Halimeda* plants studied, consisted of a lower number of segments (3–12; average 6.3), and therefore only four position categories were defined, compared to our previous study that distinguished five positions (*Neustupa & Nemcova, 2022*).

*Rindi et al. (2020)* have shown that all specimens collected from widely scattered locations in the Adriatic Sea share a common haplotype, so we may assume that all our plants sampled on the depth gradient at Cape Madona NM are genetically identical. Shape plasticity of segments within individual thalli indicates the extent of the reaction norm; a range of phenotypes developed by a single genotype over an array of environments, *i.e.,* at different depth-levels. However, the depth factor did not contribute significantly to the observed *H. tuna* segments shape and size variation. Segment shape variation spanned by the first two principal components resembled very closely the results of similar analyses on a larger set of segments (982) from two nearby sublittoral populations (*Neustupa & Nemcova, 2018*) and on a smaller set of segments (535) from distant regions on the Adriatic coast (*Neustupa & Nemcova, 2022*). This is a striking example of the deeply conserved pattern of segment shape plasticity in *H. tuna* applied independently of environmental conditions or location (*e.g.,* latitude). The shape dynamics of segments captured by PC1 was correlated to their position within the thallus, size, asymmetry, and calcium carbonate content (Fig. 2). In this study, we confirm that segments located in the basal part of the

thallus were usually smaller, more asymmetric, less calcified, and of inversely conical shape. On the other hand, segments located in the upper part of the thallus were often larger, more symmetrical, more calcified, and of oval to reniform shape (summarized in Fig. 4). *Verbruggen, De Clerck & Coppejans (2005)* referred to the basal aberrant and apical non-calcified segments as ''deviant'' and suggested that they should be excluded from the morphometric analysis of shape used to distinguish *Halimeda* species.

By adjusting size, shape, and resource allocation, macroalgae have a distinct response to light quality and quantity, which is usually correlated with depth. In *Halimeda,* the ability to modify morphology may act as an adaptive mechanism to respond to changing environmental conditions (*Yniguez, McManus & Collado-Vides, 2010*). This can be achieved by altering the overall thallus morphology. In *H. tuna*, *H. discoidea*, *H. opuntia,* and *H. macroloba,* shallow, highly irradiated localities were inhabited by compact, denser branching thalli, whereas in deeper sites with low light intensity brittle, lax branching thalli prevailed (*Bandeira-Pedrosa, Pereira & Oliveira, 2004*; *El-Manawy & Shafik, 2008*; *Pongparadon, Zuccarello & Prathep, 2017*; *Pongparadon, Nooek & Prathep, 2020*). Similarly, the thalli of shallow (4–7 m) growing *H. tuna* in Florida Keys reef systems had fewer segments than those that grew deeper (15–22 m; *Vroom et al., 2003*).

In general, thalli growing in shallow, well-irradiated habitats developed smaller segments than those growing in deeper, shaded localities (*Bandeira-Pedrosa, Pereira & Oliveira, 2004*; *Pongparadon, Zuccarello & Prathep, 2017*; *Pongparadon, Nooek & Prathep, 2020*; *Vroom et al., 2003*). It has been suggested that larger segments help plants increase their thallus height to improve light exposure (*Pongparadon, Nooek & Prathep, 2020*), or that the greater surface area of the segments may accommodate more chloroplasts in the surface utriculi and thus increase photosynthetic efficiency (*Vroom et al., 2003*). In some species, pronounced shape change was observed as a function of depth/light intensity. *Halimeda opuntia* segment shape changed from reniform under highly irradiated conditions to deeply trilobed in the lower thallus and tripartite shape in the upper thallus under shaded conditions (*Pongparadon, Nooek & Prathep, 2020*; see their Figs. 1C, 1D and 2C–2H). Similarly, the change in *H. macroloba* segment shape was mainly pronounced in the lower part of the thallus and included the change from ribbed reniform to deeply trilobed shape (*Pongparadon, Nooek & Prathep, 2020*). On the other hand, only slight variation in segment shape was observed in *H. monile* (*El-Manawy & Shafik, 2008*). In our study of *H. tuna* at Cape Madona NM in Slovenia, no variation in segment shape and size was observed as a function of depth. This could be due to a species-specific reaction norm, or our depth gradient was too short (0.5–9.2 m). The rocky bottom at Cape Madona NM reaches the depth of 12 m, and we were able to localize the deepest *H. tuna* populations at 9.2 m.

Out of the factors examined within this study, only calcium carbonate content was found to be significantly influenced by depth. In general, the average $CaCO_3$ content of 60.3% agrees with the values obtained by *Neustupa & Nemcova (2022)* who investigated upper sublittoral *H. tuna* populations over four regions on the latitudinal gradient of the Adriatic Sea (61.6%). Within other investigated Mediterranean Sea populations of this species, lower calcium carbonate content (59.7%) was found by *Prát & Hamáčková (1946)* and *Ballesteros (1991)* (45.7%). On the other hand, slightly higher values (68.1%)

were also recorded (*Bilgin & Ertan, 2002*). Only a few studies compared the degree of *Halimeda* spp. calcification between shallow and deep localities. However, the deep and shallow sites were often not within the same region, depth was not explicitly reported, and the extent of calcification was just visually assessed, making the comparison with our results difficult. In general, more heavily calcified plants were found at deeper localities, including *H. tuna* populations from Jamaica (*Goreau, 1963*; *Böhm, 1973*) and Florida (*Vroom et al., 2003*). Conversely, *Bandeira-Pedrosa, Pereira & Oliveira (2004)* visually evaluated the deep *H. tuna* populations as less calcified. The detailed study in Florida Keys reef systems revealed that *H. tuna* $CaCO_3$ content differed not only between deep (15–22 m; 79.0% in 1997 and 82.9% in 2000) and shallow (4–7 m; 75.0% in 1997 and 75.3% in 2000) populations, but also between the sampling years (*Vroom et al., 2003*). Within our investigation, we found the difference in calcium carbonate content between the shallowest and deepest populations to be comparable to the above-mentioned study. However, our depth gradient was much shorter (0.5–9.2 m). As the *H. tuna* plants growing at the deepest locality (9.2 m) had high $CaCO_3$ content, we assume that they were not light-limited. The increased $CaCO_3$ content in the deep plants may further enhance the light harvesting ability by reflecting photons through the segment (*Vroom et al., 2003*). The lower calcium carbonate content of *H. tuna* individuals at the shallowest locations (0.5 m) in this study could be due to the extremely high irradiation during sunny days, which may result in photoinhibition and lower calcification (*Häder et al., 1996*; *Beach et al., 2003*; *Vroom et al., 2003*). An alternative explanation is inhibition of carbonic anhydrase, an enzyme that catalyzes the interconversion between carbon dioxide and bicarbonate, resulting in a strong reduction in gross photosynthesis at higher light intensities (*De Beer & Larkum, 2001*).

Depth was well correlated with light intensity in our study. However, other gradients (*e.g.*, nutrient availability, extent of disturbance, seasonal temperature stratification, and herbivore pressure) should also be considered. The stable summer temperature stratification in the Gulf of Trieste could be broken by a severe bora wind when warm surface water mixes with a larger quantity of cold intermediate and deep water. Such bora events, typical of the winter season, are not uncommon even in summer (*Crise, Querin & Malačič, 2006*), when the growth of new *Halimeda* segments is maximal. As the only major fish grazer on algae, *Sarpa salpa* Linnaeus, does not usually thrive in calcified thalli (*Ballesteros, 2006*; *Orlando-Bonaca, Pitacco & Lipej, 2021*), we do not expect grazing pressure on *Halimeda* to be very high. However, sporadic fish bites were observed on the sampled segments, regardless of the depth from which they were taken.

Besides the general finding that calcium content increases with depth, we were able to evaluate how individual segments contribute to this relationship. While the apical (in the "D" position) and subapical ("C" position) segments were the main drivers of this relationship, the calcium carbonate content in the segments located basally on the thallus ("A" and "B" positions) did not change significantly with depth. We hypothesize that this is due to the spatial organization of the filaments in the basal segments, which are tightly interwoven rather than bulging into utriculi, leaving less physical space (smaller inter-utricular space) for calcification. However, this hypothesis would need to be tested by careful transmission electron microscopy studies.

## CONCLUSIONS

In summary, we found that shape, size, or asymmetry of *H. tuna* segments were not significantly affected by depth. On the other hand, plants growing deeper were more calcified. We also detected slight changes in mineral content at the microscale of the segments. We found that apical and subapical segments contributed to the $CaCO_3$ increase of mineral content at the deeper localities, whereas basal segments did not. Congruently with our previous studies, we confirmed that reniform or oval segments positioned apically or subapically play a key role in calcification of *H. tuna* in Mediterranean ecosystems.

## ACKNOWLEDGEMENTS

We thank the Marine Biology Station Piran (National Institute of Biology, Slovenia) for providing infrastructure and housing for JN and YN during our field work. Special thanks are due to Milijan Šiško for his valuable help during the fieldwork.

### Funding

This work was supported by the institutional grant of the Charles University, Prague 'Cooperatio Biology' and the Slovenian Research Agency (research core funding No. P1-0237). The funders had no role in study design, data collection and analysis, decision to publish, or preparation of the manuscript.

### Grant Disclosures

The following grant information was disclosed by the authors:
The institutional grant of the Charles University.
The Slovenian Research Agency: P1-0237.

### Competing Interests

The authors declare there are no competing interests.

### Author Contributions

- Yvonne Nemcova conceived and designed the experiments, performed the experiments, prepared figures and/or tables, authored or reviewed drafts of the article, and approved the final draft.
- Martina Orlando-Bonaca performed the experiments, authored or reviewed drafts of the article, and approved the final draft.
- Jiri Neustupa conceived and designed the experiments, analyzed the data, prepared figures and/or tables, authored or reviewed drafts of the article, and approved the final draft.

### Data Availability

The data is available at Zenodo: Nemcova Yvonne, Orlando-Bonaca Martina, & Neustupa Jiri. (2022). *Halimeda tuna* (Bryopsidales, Ulvophyceae) calcification on the depth transect in the northern Adriatic Sea; carbonate production on the microscale of individual segments [Data set]. Zenodo. https://doi.org/10.5281/zenodo.7574157.

## Supplemental Information

Supplemental information for this article can be found online at http://dx.doi.org/10.7717/peerj.15061#supplemental-information.

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
