# Peer review of "Halimeda tuna (Bryopsidales, Ulvophyceae) calcification on the depth transect in the northern Adriatic Sea; carbonate production on the microscale of individual segments"

_PeerJ, doi:10.7717/peerj.15061_

## Round 0.1 · original submission · Major Revisions

I am sorry for the substantial delay in getting a decision back to you, but I have been trying to get the final referee to send in their review. Unfortunately, they changed their mind after accepting the review request and have failed to complete their review despite numerous reminders and requests. The two referees who did respond are both encouraging about the value of the manuscript, with a number of suggestions for potential improvement. The one major issue is that one referee questions whether the differences in CaCO3 reported may be a simple function of surface area, but there is no data on surface area presented in the manuscript. The addition of data is major revision in my view, although it seems that should be simple to address and an important addition to the manuscript moving forward. Beyond that issue, the rest of the comments seem relatively straightforward to assist with clarity and terminology, and I do not expect you to have much trouble addressing them.

If you decide to incorporate the referee feedback as suggested, I ask that you provide a detailed cover letter outlining your response to each of the referee’s comments. I also point out that it is a common mistake made by many authors to respond to referee feedback in the letter only and not incorporate that same information into the manuscript itself. Future readers cannot see the response to the referees, and likely have similar questions, so we ask that your responses be reflected in the manuscript as well as the rebuttal letter. I look forward to seeing your responses and your revised manuscript.

Reviewer 1 ·

Basic reporting

The manuscript by Nemcova et al about calcification in Halimeda is an original contribution to our understanding of the response of algal calcification to major controls such as light. It explores the degree of environmental vs. species-specific control driving Halimeda article shape, size and calcification, with interesting fallout over sever reserach topics, including global change, biogeochemistry and (paleo)ecology. The paper is well written in a clear scientific language. It clearly states its aims and provides coherent results.
Some issues: some terms are misapplied and may convey confusion about the ecological role of Halimeda. Also the cited literature is biased toward tropical coral reef environments, which does not occur in the Mediterranean. I recommend to avoid the terms bioconstructor or bio-concretion while referring to Halimeda, and to add some appropriate references, as suggested in the annotated manuscript.

Experimental design

The reserach is coherent with the aims and scope of the journal. Methods and statistical treatment are correct and clearly explained

Validity of the findings

Data are robust and statistically sound, leading to well stated conclusions. Conclusions are well stated, linked to original research question and limited to supporting results

Additional comments

None

Annotated reviews are not available for download in order to protect the identity of reviewers who chose to remain anonymous.

Reviewer 2 ·

Basic reporting

The paper aim to investigate the relationship between shape, CaCO3 and depth of Halimeda tuna. The work is interesting because it shows how specimens living in deeper environment have more CaCO3.
The paper is clear, even though there are some issues with terminology, specifically with the use of the word bioconstructor. Even though Halimeda spp. are important for the carbon cycle, they cannot be called bio constructor (see comments in L:18) which is a terminology used quite often during the introduction.
In the manuscript several different shapes are mentioned. The main two were reniform and conical but other shapes (L:358) such as trilobed and tripartite were also mentioned in the discussion. While some of them are easy to imagine I strongly suggest adding a panel in figure 1 with the different shapes but facilitate the reading.
The material and methods are overall well explained even though some part need a bit more clarity (see Line:174 as an example). In the sampling section there is a part that could be easily skipped because the information (even though interesting) is not needed for the work carried out (see specific comments).
What it is not clear is why the authors chose to look at the shape ignoring the surface area which is the most likely reason why the reniform shapes have more CaCO3 stored compared to the conical ones. It would be useful to see the surface area displayed as well.

Experimental design

no comment

Validity of the findings

no comment

Additional comments

Specific comments:
L18: I would delete the phrase “..of the most important bioconstructor”. The word bioconstructor implies the persistence of the structure after dead which is not the case for the Halimeda. The phrase can be changed with “…an important biogenic structure” or “is an important contributor to the carbonate budget”

L19: change “belong among” to “.. is among..”

L37_L38: the statement “ …are major contributors to inter-reef carbonate deposits..” needs a reference.

L41-L43: the statement needs a reference.

L44: the term bioconstructor is not correct (see explanation in L18)
L78_L79: the phrase “..were suggested.” is not really clear. They were suggested as reason for the calcification process to take place?”

L85_L95: I would remove the first statement “shape of segment are species specific” because the paragraph them proceed to explain that they are plastic (depending on stressors) and good predictors of species membership (so not really species specific)

L:152 to L 159: I don’t see the relevance of this section for the paper.

L158: This phrase is already written in the introduction. Please delete it.

L174: This is not clear to me “… dried until constant weight was achieved…”. How did you do that? Usually there are a set number of hours.

L351: This is also likely related to currents and waves (stronger water motions in the surface area

L388: I do agree with the authors that considering the depth taken into consideration they are not light limited; however, I also think that the reduce calcification in the shallow might not strictly related to photoinhibition. Is there any physiological paper looking at the saturating light and calcification on Halimeda?
L358: there are several shapes in this manuscript some of them are easier to “imagine” then other. I suggest adding a panel in Figure 1 with all the shapes mentioned in the manuscript

The figure 3 is not explained properly and doesn’t add much to the paper. I would suggest giving more info in the figure caption or remove it.

Annotated reviews are not available for download in order to protect the identity of reviewers who chose to remain anonymous.

---

## Round 0.2 · accepted · Accept

I have now heard back from the more critical referee that they are satisfied with the revisions, and see no reason for additional review. As such, I am happy to move your manuscript forward. Congratulations, and thank you for selecting PeerJ as the outlet for your work.